# Validation and repurposing of the MSL-COVID-19 score for prediction of severe COVID-19 using simple clinical predictors in a triage setting: The Nutri-CoV score

Omar Yaxmehen Bello-Chavolla[1,2☯], Neftali E. Antonio-Villa[1,3☯], Edgar Ortiz-Brizuela[4], Arsenio Vargas-Vázquez[1,3], María Fernanda González-Lara[4], Alfredo Ponce de Leon[4], José Sifuentes-Osornio[4,5], Carlos A. Aguilar-Salinas[1,6,7] *

1 Unidad de Investigación de Enfermedades Metabólicas, Instituto Nacional de Ciencias Médicas y Nutrición Salvador Zubirán, Ciudad de México, Mexico, 2 División de Investigación, Instituto Nacional de Geriatría, Ciudad de México, Mexico, 3 PECEM, Faculty of Medicine, National Autonomous University of Mexico, Ciudad de México, Mexico, 4 Departamento de Infectologia, Instituto Nacional de Ciencias Médicas y Nutrición Salvador Zubirán, Ciudad de México, Mexico, 5 Direccion de Medicina, Instituto Nacional de Ciencias Médicas y Nutrición Salvador Zubirán, Ciudad de México, Mexico, 6 Direccion de Nutrición, Instituto Nacional de Ciencias Médicas y Nutrición Salvador Zubirán, Ciudad de México, Mexico, 7 Tecnologico de Monterrey, Escuela de Medicina y Ciencias de la Salud, Zapopan, Mexico

☯ These authors contributed equally to this work.
* caguilarsalinas@yahoo.com

**Data Availability Statement:** Code for verification of all analytical procedures is available at https://

## Abstract

### Background

During the COVID-19 pandemic, risk stratification has been used to decide patient eligibility for inpatient, critical and domiciliary care. Here, we sought to validate the MSL-COVID-19 score, originally developed to predict COVID-19 mortality in Mexicans. Also, an adaptation of the formula is proposed for the prediction of COVID-19 severity in a triage setting (Nutri-CoV).

### Methods

We included patients evaluated from March 16th to August 17th, 2020 at the Instituto Nacional de Ciencias Médicas y Nutrición, defining severe COVID-19 as a composite of death, ICU admission or requirement for intubation (n = 3,007). We validated MSL-COVID-19 for prediction of mortality and severe disease. Using Elastic Net Cox regression, we trained (n = 1,831) and validated (n = 1,176) a model for prediction of severe COVID-19 using MSL-COVID-19 along with clinical assessments obtained at a triage setting.

### Results

The variables included in MSL-COVID-19 are: pneumonia, early onset type 2 diabetes, age > 65 years, chronic kidney disease, any form of immunosuppression, COPD, obesity, diabetes, and age <40 years. MSL-COVID-19 had good performance to predict COVID-19 mortality (c-statistic = 0.722, 95%CI 0.690–0.753) and severity (c-statistic = 0.777, 95%CI 0.753–0.801). The Nutri-CoV score includes the MSL-COVID-19 plus respiratory rate, and

github.com/oyaxbell/nutri_cov. Data cannot be made publicly available due to ethical restrictions, however they are available on request from the Institutional Data Access Ethics Committee with contact via the corresponding author or the president of the ethics committee at arturo. galindof@incmnsz.mx, for researchers who meet the criteria for access to confidential data. For more information please consult: https://www.incmnsz. mx/opencms/contenido/investigacion/comiteEtica/miembros_comite.html.

**Funding:** The author(s) received no specific funding for this work.

**Competing interests:** The authors have declared that no competing interests exist.

pulse oximetry. This tool had better performance in both training (c-statistic = 0.797, 95%CI 0.765–0.826) and validation cohorts (c-statistic = 0.772, 95%CI 0.0.745–0.800) compared to other severity scores.

## Conclusions

MSL-COVID-19 predicts inpatient COVID-19 lethality. The Nutri-CoV score is an adaptation of MSL-COVID-19 to be used in a triage environment. Both scores have been deployed as web-based tools for clinical use in a triage setting.

## Introduction

The pandemic caused by the SARS-CoV2 virus, which is causative of COVID-19, has led to increased morbidity and mortality, posing challenges to healthcare systems worldwide [1]. Since its arrival in Mexico in late February 2020 to date, it has caused over 800,000 cases and more than 90,000 deaths attributable to COVID-19 [2]. Although most of these cases will remain mild or moderate, a group of patients could develop a severe to critical form of COVID-19 which will require quick medical assessment to prevent adverse clinical outcomes [3]. Most of these cases with severe to critical disease have underlying chronic cardiometabolic comorbidities (e.g., obesity, type 2 diabetes, arterial hypertension), chronic kidney disease, pulmonary obstructive disease and immunosuppression of any cause [4]. The high prevalence of obesity, type 2 diabetes and other chronic diseases, in addition with socioeconomic disparities, unequal access to prompt medical attention, and a lack of sufficient health infrastructure pose intense challenges in the public health care system in Mexico [5, 6]. As the disease continues to spread locally, the implementation of clinical tools to assess those patients with a higher risk for complications and lethality are needed [7, 8]. Recently, some predictive scores have been proposed and validated for COVID-19 in a hospital care setting, especially to predict critical illness hospitalization, intense care unit (ICU) admission, and mechanical ventilation support. However, these scores include specialized image acquisition or laboratories, which might not be available at most primary care scenarios in Mexico and elsewhere [9–11]. Recently, our group developed a novel mechanistic score for lethality attributable to COVID-19 (MSL-COVID-19) using age, self-reported comorbidities, and clinically suspected pneumonia, which performed adequately in a real-world scenario [12]. This tool considers the major contribution of chronic diseases to develop severe forms of COVID-19. Hence, the objective of the present study is to validate the MLS-COVID-19 to predict relevant hospitalization outcomes. Furthermore, we sought to develop an improved score based in the MLS-COVID-19 to predict disease severity including simple clinical measurements obtained in a triage setting.

## Methods

### Source of data and study population

This is a secondary analysis of the registry data of COVID-19 patients attended at the Instituto Nacional de Ciencias Médicas y Nutrición Salvador Zubirán (INMCNSZ). Briefly, we included patients aged >18 years with complete clinical data from March 16[th] to August 17[th], 2020 who were evaluated at triage at INCMNSZ, a COVID-19 reference center in Mexico City, and had confirmed SARS-CoV-2 infection by RT-PCR test in respiratory samples. All patients were followed up until September 15[th], 2020. All clinical procedures and measurements were approved

by the INCMNSZ Research and Ethics Committee, written informed consent was waived due to the observational nature of the study. Amongst all evaluated patients within the study period, we considered consecutive patients with complete data to estimate the MSL-COVID-19 score (n = 3,007). This report adheres to the TRIPOD guidelines for development and validation of predictive models.

## Outcome and timepoint setting

Clinical recovery was defined as hospital discharge based on the absence of clinical symptoms requiring inpatient management. ICU requirement was based on clinical judgment of the attending physician. Severe COVID-19 was determined as a composite event of either death, ICU admission requirement or mechanical ventilation; we estimated performance of metrics for this outcome at 7, 10, 15, 20 and 30 days, with a main focus being at 15 days given previous research indicating the disease course of mild COVID-19 cases. Follow-up time was estimated from date of symptom onset, considered as time zero of follow-up, up to last follow-up (censoring) or the composite event of severe COVID-19, which included death, requirement for invasive ventilation or ICU admission.

## Predictor variables

Information was collected prospectively at the time of triage and emergency department evaluation; candidate predictor variables included predictors which were routinely measured in a triage setting without laboratory measurements. We only considered predictors available in >80% or participants. Predictors included demographic variables including age and sex, medical history of comorbidities including type 2 diabetes, obesity, chronic obstructive pulmonary disease (COPD), asthma, hypertension, immunosuppression, HIV, cardiovascular disease (CVD), chronic kidney disease (CKD), chronic liver disease (CLD), smoking habits, and current symptoms, as described elsewhere [13]. Physical examination included weight (measured in kilograms) and height (measured in meters) to estimate the body-mass index (BMI), vital signs including pulse oximetry ($SpO_2$), respiratory rate (RR), heart rate (HR) and arterial blood pressure (BP). The Charlson Comorbidity Index (CCI), the National Early Warning Score (NEWS, NEWS2), and the quick Sequential Organ Failure Assessment (qSOFA) were also estimated to predict risk of severe COVID-19 [11, 14, 15]. Using this information, we also calculated the Mechanistic Score for COVID-19 Lethality (MSL-COVID-19), developed for prediction of COVID-19 case lethality in Mexicans using nation-wide case data (5).

## Missing data

We included complete-case analysis for validation of MSL-COVID-19 under the assumption of data missing completely at random. Next, we analyzed patterns of missing data on clinical predictors to assess patterns of missing data and performed multiple imputation for variables assumed to be completely missing at random or missing at random using multivariable imputation with chained equations within the *mice* R package, generating five multiply imputed datasets and pooling the results of modeling using Rubin's rules. Matrices of missing data are fully presented in S1 File.

## Sample size estimation

Sample size was estimated for a time-to-event model using the guidelines proposed by Riley et al implemented with the *pmsamplesize* R package [16]. We considered the prior estimated c-statistic for the MSL-COVID-19 of 0.823 and the $R^2_{CS}$ of 0.167, for an approximate event rate

of 15% within the population, which accounted for 15-day risk during a mean 11.7 days of follow-up would represent 10 cases per 1,000 patient-days or an event rate of 0.010. Using all 35 candidate predictors, the minimum sample size required for new model development was estimated to be 1,706 with 19,994.3 person-days and 200 outcome events, for a minimum of 5.71 events per predictor parameter.

## Statistical analysis

**Outcome assessment.**   We compared patients with and without Severe COVID-19 using chi-squared tests for categorical variables and Student's t-test or Mann-Whitney's U for continuous variables depending on variable distribution. We evaluated each variable for prediction of mortality, ICU admission, requirement of mechanical invasive ventilation and/or severe disease using Kaplan-Meier curves and Cox Proportional Hazard Regression models; model assumptions were verified using Schöenfeld residuals. A multivariable model was fitted to assess for predictor independence with model selection being carried out using minimizations in the Bayesian Information Criterion (BIC).

**MSL-COVID-19 model validation and predictors of severe disease.**   To validate the MSL-COVID-19 score we used Cox Proportional Hazard regression models on the continuous score and assessed model performance using Harrel's c-statistic, Sommer's $D_{xy}$ score and the calibration slope in the overall sample for prediction of mortality, ICU and/or invasive ventilation requirement and severe/critical COVID-19. To estimate the optimism in these metrics, we carried out bootstrapping (B = 1,000) using the *rms* R package, using bias corrected accelerated (BCa) bootstrapping.

**Derivation of the Nutri-CoV score.**   To increase precision of the MSL-COVID-19 score, we developed a two-step model whereby the score would be used as a first step and a second step would consider the previous risk and update it with clinical information. We included patients with complete clinical and physical examination data (n = 3,007) and split it into a training and validation datasets for patients admitted up to June 4[th] 2020 (training) and from that date up to August 17[th] (validation). To address overfitting and improve generalizability of our findings, we used Elastic Net Cox Regression, a regularization algorithm which handles multicollinearity and perform variable selection to increase generalizability using a λ penalization parameter and an α mixture parameter. We fitted an Elastic Net Cox Proportional Risk Regression Model using the both the *MAMI* and *gmlnet* R packages to incorporate coefficient estimation from multiply imputed data including al predictor variables depicted in Table 1; the ideal λ penalization parameter was estimated using k-fold cross-validation (k = 10) in the training dataset (S1 File). We selected the optimum alpha mixture parameter using simultaneous cross-validation for consecutive alpha mixture values ranging from 0 to 1 using 0.1 increments across each multiply imputed dataset and obtaining an average from all alpha values; to implement this, we used the *cva.glmnet* function of the *glmnetUtils* R package. We explored models including non-linear terms using restricted cubic splines, deciding the number of knots based on BIC minimization, and categorized models. We compared the performance of models with non-linear terms and categorized variables using cross-validation; since we observed only marginal decreases in performance with categorized variables, we proceeded with models using categorized predictors for simplification. Model coefficients from the Elastic Net Model were normalized to its ratio with the lowest absolute β coefficient to develop a point system and categories were developed to maximize separation assessed using Kaplan-Meier curves in the training dataset.

**Comparison of Nutri-CoV with other severity measures.**   We assessed the performance of the Nutri-CoV score in both the training and validation cohorts using a simple imputed

**Table 1. Patient characteristics evaluated at triage or evaluation at the emergency department at our institution comparing severe and non-severe COVID-19 cases.**

| Parameter | Non-severe COVID-19 (n = 2,432) | Severe COVID-19 (n = 575) | p-value |
|---|---|---|---|
| Age (years) | 44 (33–55) | 56 (47–66) | <0.001 |
| Male sex (%) | 1,227 (50.5%) | 403 (70.1%) | <0.001 |
| Time from symptom onset (Days) | 5.0 (3.0–8.0) | 7.0 (5.0–10.0) | <0.001 |
| BMI (kg/m2) | 28.76±5.31 | 30.53±5.96 | <0.001 |
| SBP (mmHg) | 123.98±19.10 | 126.60±17.32 | 0.003 |
| DBP (mmHg) | 77.62±12.00 | 74.47±12.56 | <0.001 |
| SpO2 (%) | 89.65±7.41 | 68.31±15.97 | <0.001 |
| Temperature (˚C) | 36.76±0.75 | 36.99±0.76 | <0.001 |
| Heart rate (bpm) | 100.48±16.96 | 103.45±18.46 | 0.003 |
| Respiratory rate (bpm) | 25.73±6.25 | 32.21±8.55 | <0.001 |
| Hospitalization | 357 (32.8%) | 237 (100.0%) | <0.001 |
| Obesity (%) | 712 (29.3%) | 268 (46.6%) | <0.001 |
| Overweight (%) | 629 (32.6%) | 215 (38.2%) | <0.001 |
| Diabetes (%) | 387 (15.9%) | 193 (33.6%) | <0.001 |
| COPD (%) | 21 (0.9%) | 10 (1.7%) | 0.101 |
| Asthma (%) | 30 (3.1%) | 2 (0.8%) | 0.068 |
| Immunosupression (%) | 92 (3.8%) | 33 (5.7%) | 0.045 |
| Hypertension (%) | 440 (18.1%) | 213 (37.1%) | <0.001 |
| HIV (%) | 24 (1.0%) | 3 (0.52%) | 0.413 |
| CVD (%) | 71 (2.9%) | 30 (5.2%) | 0.008 |
| CKD (%) | 46 (1.8%) | 16 (2.8%) | 0.782 |
| Fever (%) | 1,784 (73.4%) | 495 (86.1%) | <0.001 |
| Cough (%) | 2,006 (82.5%) | 520 (90.4%) | 0.001 |
| Headache (%) | 1,909 (78.7%) | 418 (73.1%) | 0.004 |
| Dyspnea (%) | 901 (37.0%) | 512 (89.0%) | <0.001 |
| Diarrhea (%) | 757 (31.8%) | 154 (29.6%) | 0.345 |
| Chest pain (%) | 726 (30.5%) | 211 (37.9%) | <0.001 |
| Myalgias (%) | 1,512 (63.6%) | 356 (64.0%) | 0.833 |
| Arthralgias (%) | 1,390 (58.4%) | 335 (60.5%) | 0.399 |
| Malaise (%) | 1,795 (73.8%) | 457 (79.5%) | 0.005 |
| Rhinorrhea (%) | 929 (39.1%) | 156 (28.4%) | <0.001 |
| Vomiting (%) | 260 (10.9%) | 73 (13.3%) | 0.137 |
| Abdominal pain (%) | 147 (15.0%) | 34 (14.4%) | 0.809 |
| Conjunctivitis (%) | 477 (20.1%) | 72 (13.1%) | 0.001 |
| Cyanosis (%) | 51 (2.2%) | 50 (9.15%) | <0.001 |
| qSOFA | 1.0 (0.0–1.0) | 1.0 (1.0–1.0) | <0.001 |
| NEWS | 8.0 (7.0–9.0) | 9.0 (8.0–10.0) | <0.001 |
| ABC-GOALS | 3.0 (1.0–5.0) | 6.0 (5.0–8.0) | <0.001 |
| MSL-COVID-19 | 0.0 (-5.0–4.0) | 8.0 (7.0–9.0) | <0.001 |

*Abbreviations*: BMI: Body-mass index; SBP: Systolic blood pressure; DBP: Diastolic blood pressure; SpO2: pulse oximetry; COPD: Chronic Obstructive Pulmonary Disease; HIV: Human Immunodeficiency Virus; CVD: Cardiovascular disease; CKD: Chronic Kidney Disease

dataset with *mice and rms* R packages, with correction for overoptimism being estimated using bias corrected accelerated (BCa) bootstrapping. We also estimated additional performance metrics including time-dependent sensitivity, specificity, positive and negative predictive values were obtained using the *timeROC* R packages using Inverse Probability of Censoring

Weighting (IPCW) estimation of Cumulative/Dynamic time-dependent ROC curves for 7, 10, 15, 20 and 30 days after symptom onset with weighting performed using Cox proportional risk regression. Finally, we compared performance of MSL-COVID-19, ABC-GOALS, qSOFA, NEWS, NEWS2, and the CCI to predict disease severity using decision curve analyses estimated with the *rmda* R package. A p-value <0.05 was considered as statistical significance threshold. All analyses were performed using R software version 3.6.2. To facilitate the use of the score, we implemented both the MSL-COVID-19 and the Nutri-CoV score in a web-based tool using the *ShinyApps* R package hosted at: https://uiem.shinyapps.io/nutri_cov/, and also available in Spanish at https://uiem.shinyapps.io/nutri_cov_es/.

## Sensitivity analyses

We performed recalculation of validation metrics using the *rms* package under the following scenarios: 1) Comparing participants with and without comorbidities, 2) excluding participants who were not yet discharged and were censored at the end of follow-up, 3) excluding participants who had severe COVID-19 upon admission, 4) comparing cases above and below 60 years of age, and 5) considering complete case analysis vs. multiply imputed analysis.

## Results

### Study population

We included 3,007 consecutive patients evaluated during the study period. Amongst them, 1317 patients were hospitalized at INCMNSZ (43.8%), 15 were transferred after initial evaluation (0.5%) and the rest (1,675, 55.7%) were sent for ambulatory management. The median time from symptom onset to initial clinical evaluation was 6.0 days (IQR 3.0–8.0) and for hospitalization was 7.0 days (IQR 5.0–10.0). Amongst those treated at our institution, 546 required ICU admission (18.2%) and 261 (8.7%) required mechanical invasive ventilation; median time from symptom onset to ICU admission was 8.0 days (IQR 3.5–14.0). We recorded 317 deaths within this population (10.5%), and the average time from symptom onset to death was 14.0 days (9.0–19.0). When accounting for severe COVID-19, we observed 575 cases (19.1%), with an average of 15.0 days (IQR 10.0–26.0) from symptom onset to severe outcome; overall, severe cases were older, most likely male and had higher BMI, RR, HR, body temperature and lower BP and SpO2 levels. Rates of obesity, overweight and diabetes were higher amongst severe COVID-19 cases (Table 1).

### Predictors of COVID-19 disease severity

In univariate analyses using Cox proportional risk regression, we observed a higher risk of severe COVID-19 with increasing age (HR 1.024, 95%CI 1.018–1.031), less days from symptom onset to evaluation (HR 0.931, 95%CI 0.913–0.948), male sex (HR 1.258, 95%CI 1.051–1.506), higher BMI (HR 1.019, 95%CI 1.004–1.034), first (HR 1.231, 95%CI 1.157–1.309) and second restricted cubic splines of RR (HR 0.808, 95%CI 0.747–0.874), first (HR 0.992, 95%CI 0.985–0.999) and second restricted cubic splines of $SpO_2$ (HR 0.872, 95%CI 0.845–0.901), type 2 diabetes (HR 1.365, 95%CI 1.147–1.625), and hypertension history (HR 1.408, 95%CI 1.188–1.669). When analyzing symptoms, dyspnea was a predictor of severe COVID-19 (HR 2.54, 95%CI 1.940–3.333), whilst headache (HR 0.873, 95%CI 0.688–0.996) was a protective factor. In the fully adjusted models, the only independent predictors for severe COVID-19 were increasing age, and higher RR, whilst higher $SpO_2$ was a protective factor (Table 2). Inclusion of use of supplementary oxygen in the model did not attenuate the observed associations.

**Table 2. Cox proportional risk regression models for prediction of severe COVID-19 using clinical characteristics at triage evaluation.**

| Model | Parameter | β | HR (95%CI) | p-value |
|---|---|---|---|---|
| c-statistic = 0.694 (95% CI 0.668–0.721) | MSL-COVID-19 | 0.108 | 1.114 (1.088–1.140) | <0.001 |
| c-statistic = 0.790 (95%CI 0.767–0.813) | Age | 0.024 | 1.024 (1.018–1.031) | <0.001 |
| | SpO2 (%) | -0.028 | 0.972 (0.9670.977) | <0.001 |
| | Respiratory rate | 0.022 | 1.022 (1.012–1.033) | <0.001 |
| c-statistic = 0.791 (95%CI 0.768–0.813) | MSL-COVID-19 | 0.081 | 1.085 (1.059–1.111) | <0.001 |
| | SpO2 (%) | -0.026 | 0.975 (0.969–0.980) | <0.001 |
| | Respiratory rate | 0.018 | 1.019 (1.008–1.029) | <0.001 |

## Validation of the MSL-COVID-19 score in the INCMNSZ cohort

For prediction of mortality, we identified a good discriminative capacity by the continuous values of the MSL-COVID-19 score as assessed by the calibration slope, c-statistic and Sommer's $D_{xy}$ of MSL-COVID-19 (Slope = 1.010, Dxy = 443; the c-statistic was 0.722 (95%CI 0.690–0.753). For prediction of ICU admission, the MSL-COVID-19 score also had good discriminative capacity (Slope = 0.992, Dxy = 0.663) with a slightly higher c-statistic (0.832, 95%CI 0.812–0.851); for intubation, the performance was not as good compared to the other outcomes (Slope = 0.781, Dxy = 0.885, c-statistic = 0.456, 95%CI 0.369–0.542). Overall, for prediction of severe COVID-19, we observed that MSL-COVID-19, MSL-COVID-19 had a good performance Slope = 0.781, Dxy = 0.885, c-statistic = 0.456, 95%CI 0.369–0.542). When the MSL-COVID-19 score was combined with RR and $SpO_2$, prediction of severe/critical disease improved significantly, with slight overfitting, indicating the possibility of improved performance with inclusion of these metrics to improve risk assessment (Slope = 0.979, Dxy = 0.583, c-statistic = 0.793, 95%CI 0.771–0.815, Table 2).

## Derivation of the Nutri-CoV score

Next, we sought to develop a score to predict disease severity including previous assessment with MSL-COVID-19. Using this Elastic Net Cox Regression, we confirmed that a combination of RR, SpO2 and the MSL-COVID-19 score improved prediction of severe COVID-19 (Fig 1). We fitted the model in the training cohort (n = 1831, 373 outcomes) and identified an

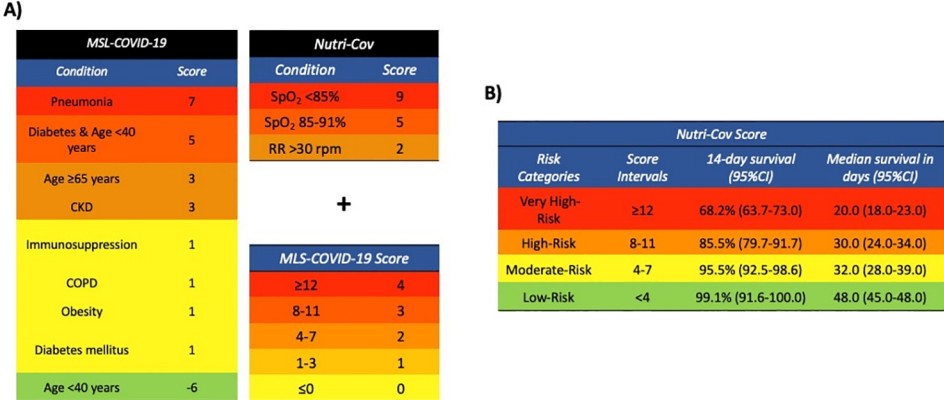

**Fig 1. Proposal for calculation of the Nutri-CoV score with initial estimation of the MSL-COVID-19 score for stratification of risk for severe COVID-19.** Abbreviations: RR, Respiratory Rate; SpO2: Oxygen saturation, CKD = Chronic Kidney Disease, COPD: Chronic Obstructive Pulmonary Disease.

adequate discriminative capacity (slope = 1.0013, Dxy = 0.523, c-statistic = 0.772, 95%CI 0.745–0.800). Using time-dependent ROC curves with IPCW obtained with the Kaplan-Meier estimator, we identified that a cut-off of 8.0 had 94.4% time-dependent sensitivity (95%CI 90.8–98.6%), 34.7% time-dependent specificity (95%CI 30.2–39.1%) to identify subjects with 15-day risk of severe COVID-19. The PPV and NPV were 22.4% (95%CI 18.9–25.9) and 96.9% (95%CI 94.7–99.1%). Using this cut-off, we observed 358 false positives and 15 false negatives; lowering the cut-off to 5 for low risk only yielded 3 false negatives and as such was chosen to represent the low-risk category in the training cohort.

### Validation of the Nutri-CoV score in the validation cohort

Next, we applied the model to the validation cohort (n = 1,176, 202 outcomes), where the score also presented a good predictive capacity (slope = 0.999, Dxy = 0.592, c-statistic = 0.797, 95%CI 0.765–0.826). With time dependent AUROC, we observed that the cut-off of 8 had had 95.0% time-dependent sensitivity (95%CI 90.4–99.5%), 37.0% time-dependent specificity (95%CI 31.0–42.9%) to identify subjects with 15-day risk of severe COVID-19. The PPV and NPV were 22.4% (95%CI 18.1–27.1) and 96.9% (95%CI 95.0–99.8%). Using the low-risk category cut-off (Nutri-CoV<5) we identified only 1 false negative (Table 3). The proposed risk cut-offs in the training cohort significantly discriminate 15-day risk of severe COVID-19 in the validation cohort using Kaplan-Meier curves (log-rank p<0.001, Fig 2).

### Comparison of Nutri-CoV with severity scores

Finally, we compared performance of Nutri-CoV compared to MSL-COVID-19, ABC-GOALS, the ROX index, qSOFA, and NEWS to predict COVID-19 severity overall and in the validation cohort (Table 4). We observed a significant and improved performance, as measures by time-dependent AUROC, for Nutri-CoV to predict 15-day risk of severe COVID-19 in comparison to all scores in the validation cohort. When assessing its performance compared to other indexes using decision curve analysis, significant clinical benefit was established for the Nutri-CoV score compared to other indexes (Fig 3).

### Sensitivity analyses

We conducted sensitivity analyses for relevant risk categories and to assess methodological decisions during score derivation (Table 5). Notably, Nutri-CoV observed improved performance in cases who were younger than age 60 and had no comorbidities, with decreased performance for use of only hospitalized cases. Notably, we observed a minimal decrease in performance when conducting complete-case analysis compared to multiply imputed data or excluding cases who had not been discharged.

## Discussion

Here, we validated the MSL-COVID-19 score for prediction of inpatient mortality at a COVID-19 reference center in Mexico City. We also demonstrated the role of MSL-COVID-19 in

**Table 3. Comparison of time-dependent sensitivity, specificity, positive and negative predictive values for Nutri-CoV in the validation cohort for prediction of severe COVID-19 at 7, 15 and 30 days using a Nutri-CoV cut-off of 8.**

| Time point | Sensitivity (95%CI) | Specificity (95%CI) | PPV (95%CI) | NPV (95%CI) |
|---|---|---|---|---|
| 7 days | 94.44 (83.86–100.0) | 68.53 (65.69–71.37) | 4.47 (2.39–6.55) | 99.87 (99.61–100.0) |
| 15 days | 94.97 (90.40–99.54) | 36.95 (30.95–42.95) | 22.58 (18.05–27.11) | 97.43 (95.04–99.82) |
| 30 days | 92.89 (88.09–97.69) | 11.36 (1.97–20.75) | 52.57 (44.26–60.88) | 60.16 (33.62–86.70) |

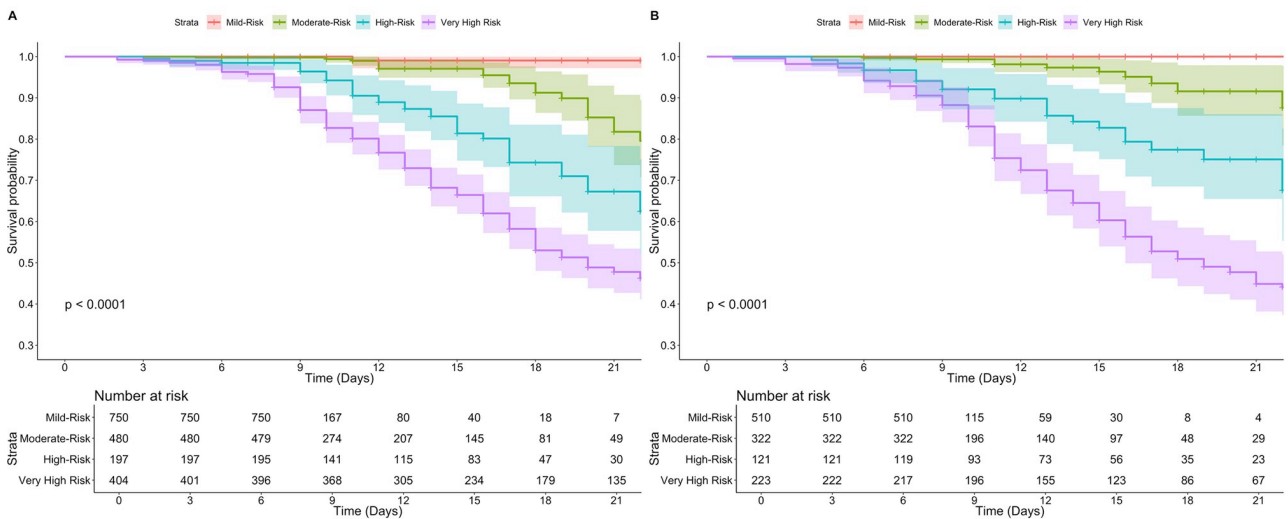

**Fig 2. Kaplan-Meier analysis of risk categories for the Nutri-CoV score according to proposed cut-offs presented in Fig 1 for the training (A) and validation cohorts (B).**

predicting severe and critical COVID-19 and how this estimation can be improved by considering additional clinical data obtained at triage. The repurposed score, which we named Nutri-CoV, includes demographics and comorbidity assessment as well as physical examination in the form of SpO2 and RR assessment, achieved significant discriminative capacity to detect potential cases of severe COVID-19. We deployed both the MSL-COVID-19 and the Nutri-CoV scores onto a web-based tool which could be readily used within a triage setting to identify individuals at the highest risk of developing COVID-19 complications. Our model could be used to make prompt decisions regarding timely admission and treatment initiation in patients at risk of severe and critical COVID-19, as well as resource allocation in the setting of increased healthcare stress during pandemic peaks. Given the remarkable increase in the availability of COVID-19 predictive models, we expect the derivation of our score will be helpful in triage settings with similarities to our institution; however, extensive external validation and calibration studies are required to evaluate its performance in a triage prior to clinical utilization [17]. Given the conducted sensitivity analyses, we do not currently recommend the use of our score for in-hospital clinical deterioration, which could be assessed with other available tools after due external validation [18].

Underlying factors which explain the pathogenesis of severe COVID-19 complications have been extensively studied [19, 20]. A role for metabolic comorbidities including type 2

**Table 4. Comparison of severity scores with the developed Nutri-CoV score in the validation cohort using time-dependent area under the ROC curves (t-AUROC) using an age and sex adjusted Cox estimator.**

| Score | c-statistic (95%CI) | 7-day t-AUROC | 15-day t-AUROC | 30-day t-AUROC |
|---|---|---|---|---|
| Nutri-CoV | 0.797 (0.765–0.829) | 0.864 | 0.914 | 0.877 |
| MSL-COVID-19 | 0.713 (0.669–0.757) | 0.769 | 0.780 | 0.998 |
| ABC-GOALS | 0.759 (0.718–0.800) | 0.844 | 0.836 | 0.856 |
| Charlson comorbidity index | 0.654(0.604–0.703) | 0.710 | 0.641 | 0.536 |
| qSOFA | 0.657 (0.629–0.685) | 0.679 | 0.752 | 0.720 |
| NEWS | 0.733 (0.696–0.770) | 0.820 | 0.855 | 0.750 |
| NEWS2 | 0.727 (0.690–0.765) | 0.817 | 0.848 | 0.972 |

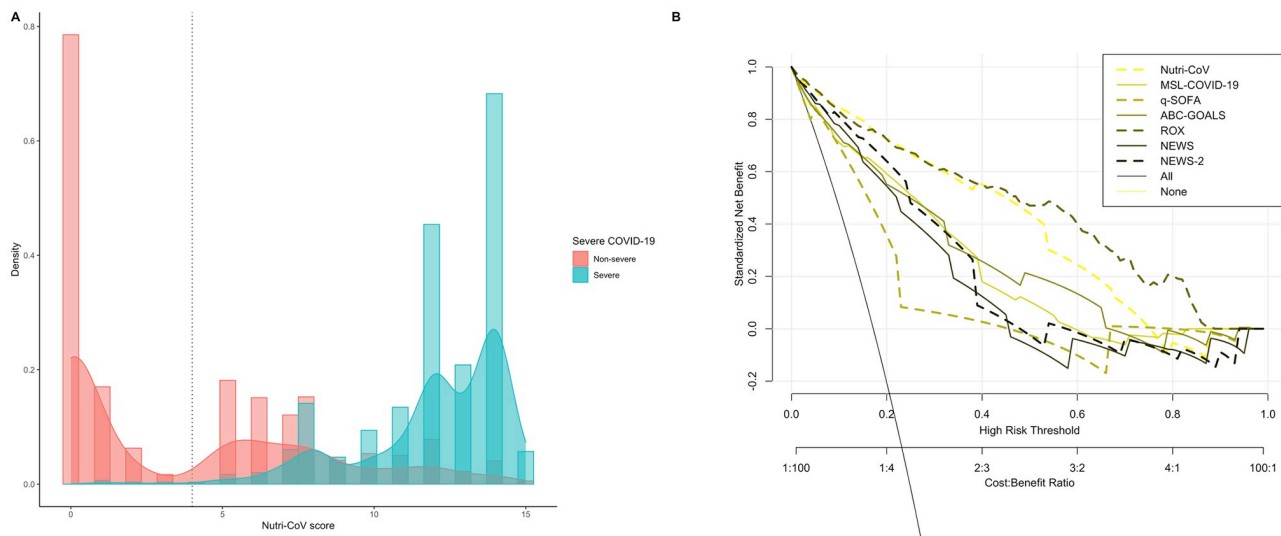

**Fig 3. Density histogram of Nutri-CoV scores with the proposed cut-off for the low risk category within the validation cohort (A) an decision curve analysis to assess the clinical benefit of Nutri-CoV compared to other severity scores including MSL-COVID-19, NEWS, NEWS-2, qSOFA, Charlson comorbidity index and ABC-GOALS.**

diabetes and its associated glycemic control, obesity, hypertension, older age and male sex has been reported as risk factors for severe COVID-19. In Mexicans, our group developed the MSL-COVID-19score, which attempts to capture the risk of most of these factors with a particular focus on predicting mortality [12]. Despite the relevance of estimating mortality risk in COVID-19, the elevated healthcare stress which has been steadily observed during the pandemic makes the consideration of ICU admission and requirement for invasive intubation relevant, especially for resource allocation and prompt treatment initiation of high-risk patients [21, 22]. However, controversy surrounding early intubation for severe COVID-19 could make identification of these cases relevant for close clinical follow-up and evaluation. Previous severity scores developed for COVID-19 consider radiological and laboratory findings in relation to clinical status and comorbidities [10]. Nevertheless, a main concern in low-resource settings is the availability of healthcare resources and a highly sensitive clinical score might be relevant to detect severe disease; whilst the MSL-COVID-19

**Table 5. Sensitivity analyses of Nurtri-CoV performance on selected groups in the validation cohort, comparing performances and calibration measures corrected for optimism using bias corrected accelerated bootstrapping (B = 1,000).**

| Subgroup | N/Outcomes | c-statistic (95%CI) | Calibration slope | Sommer's $D_{xy}$ |
|---|---|---|---|---|
| Imputed analysis | 1,176/202 | 0.797 (0.765–0.829) | 0.998 | 0.594 |
| Complete-case analysis | 1,080/196 | 0.789 (0.756–0.822) | 0.995 | 0.577 |
| >60 years | 262/94 | 0.716 (0.663–0.769) | 1.005 | 0.430 |
| ≤60 years | 914/108 | 0.831 (0.792–0.871) | 1.002 | 0.663 |
| No comorbidities | 552/36 | 0.895 (0.854–0.936) | 0.985 | 0.791 |
| ≥1 comorbidity | 624/166 | 0.735 (0.694–0.776) | 1.005 | 0.470 |
| Outcome-1 | 494/201 | 0.683 (0.643–0.724) | 1.010 | 0.366 |
| Outcome-2 | 1,079/196 | 0.789 (0.756–0.822) | 1.002 | 0.579 |

Outcome-1 = Excluding cases who were not hospitalized; Outcome-2 = Excluding cases who were not yet discharged.

score is useful for quick clinical examination, updating these estimations with current clinical status improves its predictive capacity. Nutri-CoV shows excellent discriminative capacity for low-risk cases, with a low number of eventual false positives which makes it a good screening tool to predict disease severity and resource allocation.

Onset of acute respiratory distress syndrome (ARDS) in COVID-19 patients can be attributable to underlying endothelial vascular injury which disrupts the regulation of pulmonary blood flow, leading to a ventilation-perfusion mismatch and eventually reducing oxygenation despite increased lung compliance and thrombogenesis [23]. Low oxygenation can be promptly and objectively assessed using pulse oximetry, allowing identification of COVID-19 related early hypoxia, particularly when no overt dyspnea is present [24]. This pattern of lung injury, related to decreased elastance in the setting of increased lung compliance has been described as Type L COVID pneumonia, which has different implications compared to type H COVID pneumonia, more traditionally linked to ARDS [24–26]. Hypoxemia usually leads to increase in minute ventilation by increasing tidal volume, which can be assessed as an increased respiratory rate despite absence of dyspnea in the setting or normal lung compliance. Notably, lung function is impaired in comorbidities linked to increased risk of severe COVID-19 including obesity and type 2 diabetes [27–29]; furthermore, most cardio-metabolic comorbidities have been linked to increased expression of the angiotensin converting enzyme (ACE) [30, 31]. These factors are jointly evaluated by the Nutri-CoV and MSL-COVID-19 scores, thus indicating its potential to capture pathophysiological components that eventually lead to severe COVID-19 disease and increased mortality. Notably, although dyspnea was associated with increased disease severity, it was not an independent predictor severe disease, probably because of collinearity with variables already included such as pulse oximetry and respiratory rate. The use of our algorithm should accompany clinical judgement as the evaluation of any individual component of the score does not predict the outcome with absolute certainty.

Our study had some strengths and limitations. By assessing patients who were considered to have mild, moderate and severe COVID-19, we were able to capture a wide array of clinical characteristics at triage evaluation. Nevertheless, since all patients were examined from a single center, the possibility of referral or representability bias might reduce performance of Nutri-CoV in other populations. MSL-COVID-19 was developed using nationally representative data, the good performance shown within our institution, validates its implementation in low resource settings. A potential limitation of our approach is that pulse oximetry evaluation was performed in patients who are already on supplementary oxygen; whilst this was controlled in the statistical analysis and considered within the Elastic Net regression framework, the possibility of residual confounding is present. Considering the use of a training and a validation dataset as well as cross-validation for selection of penalization parameters, the model is likely to have adequate performance; however, our model is better suited for adequate performance within our institution or similar institutions within of Mexico and thus requires additional external validation to confirm its applicability in other settings. The web application should be useful to physicians within our institution and for external settings once it is externally validated and could be complementary to decision making in a triage setting.

In conclusion, we validated the MSL-COVID-19 score using data from a COVID-19 reference center in Mexico City. We demonstrated this score could be useful to predict other outcomes related to COVID-19 including ICU admission and requirement for invasive ventilation. Including clinical parameters in the MSL-COVID-19 score, related to COVID-19 ventilatory pathophysiology, increases its performance for prediction of severe COVID-19 and makes it a useful tool for clinical decision making and resource allocation during the

healthcare stress of a pandemic scenario. Both the MSL-COVID-19 and Nutri-CoV scores were deployed as interactive webtools for clinical use in a triage setting.

## Supporting information

**S1 File.**
(DOCX)

## Acknowledgments

AVV, NEAV are enrolled at the PECEM program of the Faculty of Medicine at UNAM. NEAV and AVV are supported by CONACyT. The authors would like to acknowledge the invaluable work of all healthcare workers at the Instituto Nacional de Ciencias Médicas y Nutrición Salvador Zubirán for its community in managing the COVID-19 epidemic. Its participation in the COVID-19 surveillance program has made this work a reality, we are thankful for your effort.

## Author Contributions

**Conceptualization:** Omar Yaxmehen Bello-Chavolla, Neftali E. Antonio-Villa, Edgar Ortiz-Brizuela, Arsenio Vargas-Vázquez, María Fernanda González-Lara, Alfredo Ponce de Leon, José Sifuentes-Osornio, Carlos A. Aguilar-Salinas.

**Data curation:** Omar Yaxmehen Bello-Chavolla, Neftali E. Antonio-Villa, Edgar Ortiz-Brizuela, Arsenio Vargas-Vázquez, María Fernanda González-Lara, Alfredo Ponce de Leon, José Sifuentes-Osornio, Carlos A. Aguilar-Salinas.

**Formal analysis:** Omar Yaxmehen Bello-Chavolla, Neftali E. Antonio-Villa, Arsenio Vargas-Vázquez, Alfredo Ponce de Leon, José Sifuentes-Osornio, Carlos A. Aguilar-Salinas.

**Funding acquisition:** Alfredo Ponce de Leon, Carlos A. Aguilar-Salinas.

**Investigation:** Omar Yaxmehen Bello-Chavolla, Neftali E. Antonio-Villa, Edgar Ortiz-Brizuela, María Fernanda González-Lara, Alfredo Ponce de Leon, José Sifuentes-Osornio, Carlos A. Aguilar-Salinas.

**Methodology:** Omar Yaxmehen Bello-Chavolla, Neftali E. Antonio-Villa, Edgar Ortiz-Brizuela, Arsenio Vargas-Vázquez, María Fernanda González-Lara, Alfredo Ponce de Leon, José Sifuentes-Osornio, Carlos A. Aguilar-Salinas.

**Project administration:** Omar Yaxmehen Bello-Chavolla, Alfredo Ponce de Leon, José Sifuentes-Osornio, Carlos A. Aguilar-Salinas.

**Resources:** Edgar Ortiz-Brizuela, José Sifuentes-Osornio, Carlos A. Aguilar-Salinas.

**Software:** Omar Yaxmehen Bello-Chavolla, Carlos A. Aguilar-Salinas.

**Supervision:** Alfredo Ponce de Leon, José Sifuentes-Osornio, Carlos A. Aguilar-Salinas.

**Validation:** Omar Yaxmehen Bello-Chavolla, Arsenio Vargas-Vázquez, María Fernanda González-Lara, Carlos A. Aguilar-Salinas.

**Visualization:** Omar Yaxmehen Bello-Chavolla, Neftali E. Antonio-Villa, Arsenio Vargas-Vázquez, Carlos A. Aguilar-Salinas.

**Writing – original draft:** Omar Yaxmehen Bello-Chavolla, Neftali E. Antonio-Villa, Edgar Ortiz-Brizuela, Arsenio Vargas-Vázquez, María Fernanda González-Lara, Alfredo Ponce de Leon, José Sifuentes-Osornio, Carlos A. Aguilar-Salinas.

**Writing – review & editing:** Omar Yaxmehen Bello-Chavolla, Neftali E. Antonio-Villa, Edgar Ortiz-Brizuela, Arsenio Vargas-Vázquez, María Fernanda González-Lara, Alfredo Ponce de Leon, José Sifuentes-Osornio, Carlos A. Aguilar-Salinas.

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
