## [Decision Letter · Decision Letter 0]

28 Sep 2020

PONE-D-20-25115

Validation and repurposing of the MSL-COVID-19 score for prediction of severe COVID-19 using simple clinical predictors in a triage setting: The Nutri-CoV score

PLOS ONE

Dear Dr. Aguilar-Salinas,

Thank you for submitting your manuscript to PLOS ONE. After careful consideration, we feel that it has merit but does not fully meet PLOS ONE’s publication criteria as it currently stands. Therefore, we invite you to submit a revised version of the manuscript that addresses the points raised during the review process.

We look forward to receiving your revised manuscript.

Kind regards,

Itamar Ashkenazi

Academic Editor

PLOS ONE

Journal Requirements:

2. We note that you state in your ethics statement in the online submission form "proceedings were approved by the INCMNSZ Research and Ethics Committee". Please clarify whether your ethics committee specifically approved this study and provide the full name of this committee.

3. Please include the date(s) on which you accessed the databases or records to obtain the data used in your study.

6. Please ensure that you refer to Figure 2 in your text as, if accepted, production will need this reference to link the reader to the figure.

Reviewers' comments:

Reviewer's Responses to Questions

**Comments to the Author**

1. Is the manuscript technically sound, and do the data support the conclusions?

Reviewer #1: Partly

Reviewer #2: Yes

2. Has the statistical analysis been performed appropriately and rigorously? 

Reviewer #1: No

Reviewer #2: Yes

3. Have the authors made all data underlying the findings in their manuscript fully available?

Reviewer #1: Yes

Reviewer #2: No

4. Is the manuscript presented in an intelligible fashion and written in standard English?

Reviewer #1: Yes

Reviewer #2: Yes

5. Review Comments to the Author

Reviewer #1: I have a number of concerns regarding statistical design, analysis, and representation. I will try to go in order of line number.

Line 112. It is unclear to me whether or not the univariate analyses influenced which variables were used in the multivariable model which used BIC for model selection. For example, were only variables that were considered significantly associated with the outcome (p<0.05) used in the multivariable analysis?

Line 116-119. Was the data used to make/decide the 4 risk group stratification the same data that was used to validate it? Or was the validation set used to validate it?

Line 120. For bootstrapping, I assume the ordinary bootstrap method was used. Which confidence interval was used for the provided metrics? Percentile? BCA? etc. It is worth mentioning that the validation functions in the rms package are not intended to be used in complicated modeling designs wherein decisions such as cutoffs for continuous variables, risk stratification, etc. with the same data that is used to validate it. The result, optimistic estimates for performance. Even if you use the bootstrap later because you are not including data-driven decisions in the bootstrap process. You will likely need to code the loop yourself to include these elements in the bootstrap process.

Line 129 indicates that lambda is the tuning parameter of the Elastic-Net model, but this is only half true. Lambda is the penalty parameter in a LASSO model -- Elastic-Net adds in a mixture parameter, typically represented as alpha, that determines how much LASSO penalty and how much ridge penalty is used. Can the authors please tell me how much mix, if any between the two were used? This does not have to appear in the main paper, but some more details in an appendix or similar would be helpful. This includes how the hyperparameter optimization was done -- exhaustive grid search, randomly sampled xxx points from the hypergrid, etc.

Line 131. glmnet instead of gmlnet.

Line 132. For design, it appears that there may be a typo with regards to the number of folds, 20.

If k is indeed 20, then within the CV routine 1/20th of the sample used as the test set at any given point. This amounts to about 58 patients, approximately 6 of which will be events. In my view, this is too few. I strongly recommend that the authors instead use a 10 fold or perhaps a 5 fold CV.

Line 136. likelihood ratios instead of likelihood rations

Line 174, 178, 187. why are the c-statistics and AUC different? did you have censoring? if so, the AUC might have been calculated using a method that ignores this.

Line 190. Was the cutoff created in the training cohort only?

Line 190 - 197. Are these estimates all from the validation cohort?

Line 259 "thus increasing the external validity of our model" how? including more variables does not necessarily increase the external validity. would remove this.

Line 268. "the model is likely to have adequate performance when applied in external datasets" you have no external data to suggest this.

Missing limitation elements from discussion. For example, the study is skewed in the direction of your model because the data used to validate your model is very similar to the data used to fit/train your model. The other models under evaluation here were, if my understanding is correct, were derived using data from other institutions. If you compare how well their respective papers said they would perform and compare that to how well they performed on your data, you will likely see that their models did not perform as well.

The limitations here as I see it is three-fold: 1) repeated misuse of the bootstrap method, specifically not including cutoff and risk stratification inside the validation; 2) too many folds. i suspect that you will be better off using 5-fold or some such CV. Repeating the CV routine many times (repeated k-fold CV) would be one step better than that.; 3) recognize in the discussion that your model will inherently do better than models that were created outside of your institution.

Please also remake the plots a higher quality file as they are not legible in their current state.

Reviewer #2: In their paper " Validation and repurposing of the MSL-COVID-19 score for prediction of severe COVID-19 using simple clinical predictors in a triage setting: The Nutri-CoV score", Bello-Chavolla et al. develop prediction scores for several aspects of the COVID-19 disease using time-to-event models and present applications. In principle, the analysis is sound. However, I have one major point the authors have to address and a couple of minor comments.

Major point:

- In time-to-event models it is very important to define the time-scale exactly. Time zero is not defined in the paper. The authors write "We included patients aged >18 years with complete clinical data from March 16th to June 4th, 2020 who were evaluated at triage at the Instituto Nacional de Ciencias Médicas y Nutrición Salvador Zubirán (INCMNSZ), a COVID-19 reference center in Mexico City...". I would therefore conclude that the timepoint of presentation is taken to be timepoint zero. This would be problematic as it creates an immortal time bias for the presenting population. The interpretation of the model would then only be applicable in a setting where the same pattern of presenting at the triag unit would be present. A more logical timepoint zero would be the onset of symptoms, a population that would be well-defined. At the very least the authors need to perform a sensitivy analysis for such model, using time-dependent covariates to account for the immortal time bias

Minor points:

- Wording l. 81: "all proceedings" is somewhat unclear (all analyses?)

- l. 107: It is only meaningful to to perform the "outcome assessment" at several time horizons which need to be specified. It is not meaningful to use the data set "as is", i.e. only the presence of the outcome at any timepoint.

- l. 112: As the assumptions are checked visually, the plot need to be supplied as a supplement.

- l. 136 (and other places): The curves need to be evaluated at time horizons. These need to be clearly specified.

- Some of the predictors are known to act non-linearly such as age. As penalized regression is used, the authors are advised to include non-linear termas, such as quadratic terms.

- The authors should stress that the validation is internal and not external. The authors should also use test-data instead of validation data to avoid confusion when describing cross-validation (the cross-validation uses the validation data, evaluation is performed in the test data).

- The model should be evaluated in a stratified way using the known risk groups, as this is highly relevant for practical application.

- The authors need to provide a URL for the data set.

6. PLOS authors have the option to publish the peer review history of their article (what does this mean?). If published, this will include your full peer review and any attached files.

Reviewer #1: No

Reviewer #2: No

---

## [Author Response · Author response to Decision Letter 0]

10 Nov 2020

November 2nd, 2020

Dear Prof. Itamar Ashkenazi,

Academic Editor, PLOS ONE.

Thank you for giving us the opportunity to revise our work. We have addressed the reviewers’ invaluable comments and modified the manuscript accordingly. We believe the manuscript to be better after these revisions. Considering recent evidence on the quality of predictive modeling in COVID-19, we have rewritten several sections of our manuscript to adhere to TRIPOD guidelines and expanded our dataset to account for most recent cases, with an overall larger sample size (n=3,007). Below, we include a detailed response to each of the reviewer’s inquiries. We appreciate the opportunity to submit our work for consideration by PLoS One.

Sincerely,

Carlos A. Aguilar-Salinas, MD PhD

Corresponding author

Reviewer #1: 

I have a number of concerns regarding statistical design, analysis, and representation. I will try to go in order of line number.

R= Thank you for your revision, the points you raised helped us re-evaluate our statistical design and provide more transparent reporting, we are thankful for your suggestions.

Line 112. It is unclear to me whether or not the univariate analyses influenced which variables were used in the multivariable model which used BIC for model selection. For example, were only variables that were considered significantly associated with the outcome (p<0.05) used in the multivariable analysis?

R= We evaluated all variables presented in Table 1 for inclusion in the model, regardless of their univariate association with Severe COVID-19 to rule out potential exclusion of variables which were associated but failed to reach a significance threshold prior to adjustment. We have now adhered our reporting to the TRIPOD guidelines and have included this specification into the revised version of our manuscript. 

Line 116-119. Was the data used to make/decide the 4 risk group stratification the same data that was used to validate it? Or was the validation set used to validate it?

R= We used validation data to validate these cut-offs which were obtained in the original derivation paper for the MSL-COVID-19 score (https://doi.org/10.1210/clinem/dgaa346). For the Nutri-CoV score, we estimated risk categories within the training cohort and then evaluated in the validation cohort. 

Line 120. For bootstrapping, I assume the ordinary bootstrap method was used. Which confidence interval was used for the provided metrics? Percentile? BCA? etc. It is worth mentioning that the validation functions in the rms package are not intended to be used in complicated modeling designs wherein decisions such as cutoffs for continuous variables, risk stratification, etc. with the same data that is used to validate it. The result, optimistic estimates for performance. Even if you use the bootstrap later because you are not including data-driven decisions in the bootstrap process. You will likely need to code the loop yourself to include these elements in the bootstrap process.

R= We agree with your comments. The development and internal validation of the MSL-COVID-19 score was carried out in a separate dataset and we now validated it using data from our institution. We used bootstrapping (B=1,000) to correct for overoptimism with bias corrected accelerated boostrapping when validating all scores using the rms package; for this purpose, we only used the continuous scores and not the risk categories. Since this is an external validation of the MSL-COVID-19 model, we expect the score to have a less optimistic performance compared to the derivation dataset. We have included these methodological details into the revised version of our manuscript.

Line 129 indicates that lambda is the tuning parameter of the Elastic-Net model, but this is only half true. Lambda is the penalty parameter in a LASSO model – Elastic-Net adds in a mixture parameter, typically represented as alpha, that determines how much LASSO penalty and how much ridge penalty is used. Can the authors please tell me how much mix, if any between the two were used? This does not have to appear in the main paper, but some more details in an appendix or similar would be helpful. This includes how the hyperparameter optimization was done – exhaustive grid search, randomly sampled xxx points from the hypergrid, etc.

R= Thank you for your comment, we selected the optimum alpha mixture parameter using simultaneous cross-validation for consecutive alpha mixture values ranging from 0 to 1 using 0.1 increments in each multiply imputed dataset and obtaining an average from all alpha values; to implement this, we used the cva.glmnet function of the glmnetUtils R package. We then used the identified alpha value to fit the Elastic Net model using both MAMI and glmnet R packages to consider multiply imputed data and to obtain the Elastic Net estimator as previously described:

1. http://mami.r-forge.r-project.org/MAMI_manual.pdf

2. Schomaker, M. (2012). Shrinkage averaging estimation. Statistical Papers 53, 1015–1034. 

Line 131. Glmnet instead of gmlnet.

R= Thank you, we have corrected this typo.

Line 132. For design, it appears that there may be a typo with regards to the number of folds, 20. If k is indeed 20, then within the CV routine 1/20th of the sample used as the test set at any given point. This amounts to about 58 patients, approximately 6 of which will be events. In my view, this is too few. I strongly recommend that the authors instead use a 10-fold or perhaps a 5 fold CV.

R= We have updated the sample size for model development and re-considered the data splitting method to include model development from March 16th to June 4th and validation From June 5th to August 17th in accordance with TRIPOD guidelines. Using this technique, we ended up with a training sample of 1,831 (373 events) and a validation sample of 1176 (202 events). We performed cross-validation using k=5 and k=10, we present the results of coefficient comparison in Supplementary Material.

Line 136. likelihood ratios instead of likelihood rations

R= Thank you, this has now been corrected.

Line 174, 178, 187. Why are the c-statistics and AUC different? Did you have censoring? If so, the AUC might have been calculated using a method that ignores this.

R= Thank you for your observation. We re-estimated c-statistics and AUC to account for censoring, we are now only presenting c-statictics and its corresponding confidence intervals to assess model performance. We also assessed time-dependent ROC curves and time-dependent sensitivity, specificity, positive and negative predictive values, which were estimated using the timeROC R package.

Line 190. Was the cutoff created in the training cohort only?

R= Yes, the cut-off was created in the training cohort and subsequently validated in the validation cohort. We have included this specification in the revised version of the manuscript.

Line 190 - 197. Are these estimates all from the validation cohort?

R= We have estimated metrics for both training and validation cohorts and included them within the revised version of the manuscript. For model comparison and sensitivity analyses, we only used data from the validation cohort.

Line 259 "thus increasing the external validity of our model" how? including more variables does not necessarily increase the external validity. would remove this.

R= We agree with your observation, we have removed this comment.

Line 268. "the model is likely to have adequate performance when applied in external datasets" you have no external data to suggest this.

R= Thank you, we agree with your observation. We have removed this comment from the revised version of our manuscript.

Missing limitation elements from discussion. For example, the study is skewed in the direction of your model because the data used to validate your model is very similar to the data used to fit/train your model. The other models under evaluation here were, if my understanding is correct, were derived using data from other institutions. If you compare how well their respective papers said they would perform and compare that to how well they performed on your data, you will likely see that their models did not perform as well.

R= We agree with your assertion, we have included this within the limitations of our work in the revised version of the manuscript. The revised section reads: “however, our model is likely to perform better within our institutions or similar institutions outside of Mexico and require additional external validation to confirm its applicability in other settings”.

The limitations here as I see it is three-fold: 1) repeated misuse of the bootstrap method, specifically not including cutoff and risk stratification inside the validation.

R= To account for this, we have only validated the continuous score under the assumption that risk categories could perform differently across different populations. 

2) too many folds. i suspect that you will be better off using 5-fold or some such CV. Repeating the CV routine many times (repeated k-fold CV) would be one step better than that.

R= All models have been fit with both k-fold cross-validation, with k=5 and k=10; we present comparison of these results in Supplementary Material.

3) recognize in the discussion that your model will inherently do better than models that were created outside of your institution.

Please also remake the plots a higher quality file as they are not legible in their current state.

R= Thank you, we have now appended high quality files for all our images.

Reviewer #2: In their paper " Validation and repurposing of the MSL-COVID-19 score for prediction of severe COVID-19 using simple clinical predictors in a triage setting: The Nutri-CoV score", Bello-Chavolla et al. develop prediction scores for several aspects of the COVID-19 disease using time-to-event models and present applications. In principle, the analysis is sound. However, I have one major point the authors have to address and a couple of minor comments.

Major point:

- In time-to-event models it is very important to define the time-scale exactly. Time zero is not defined in the paper. The authors write "We included patients aged >18 years with complete clinical data from March 16th to June 4th, 2020 who were evaluated at triage at the Instituto Nacional de Ciencias Médicas y Nutrición Salvador Zubirán (INCMNSZ), a COVID-19 reference center in Mexico City...". I would therefore conclude that the timepoint of presentation is taken to be timepoint zero. This would be problematic as it creates an immortal time bias for the presenting population. The interpretation of the model would then only be applicable in a setting where the same pattern of presenting at the triag unit would be present. A more logical timepoint zero would be the onset of symptoms, a population that would be well-defined. At the very least the authors need to perform a sensitivy analysis for such model, using time-dependent covariates to account for the immortal time bias.

R= Thank you for your observation, we have included a detailed description of time zero and time estimation to reduce the likelihood of immortal time bias. We have modified the manuscript as follows: “Briefly, we included patients aged >18 years with complete clinical data from March 16th to August 17th, 2020 who were evaluated at triage at INCMNSZ, a COVID-19 reference center in Mexico City, and had confirmed SARS-CoV-2 infection by RT-PCR test in respiratory samples. All patients were followed up until September 15th, 2020 […] Follow-up time was estimated from date of symptom onset, considered as time zero of follow-up, up to last follow-up (censoring) or the composite event of severe COVID-19, which included death, requirement for invasive ventilation or ICU admission.”

Minor points:

- Wording l. 81: "all proceedings" is somewhat unclear (all analyses?)

R= We referred to all clinical procedures and measurements. We have now corrected this.

- l. 107: It is only meaningful to to perform the "outcome assessment" at several time horizons which need to be specified. It is not meaningful to use the data set "as is", i.e. only the presence of the outcome at any timepoint. - l. 136 (and other places): The curves need to be evaluated at time horizons. These need to be clearly specified.

R= We agree with your comment, we have now estimated performance of all risk metrics to estimate risk at days 7, 15, and 30-days using time dependent ROC curves.

- l. 112: As the assumptions are checked visually, the plot need to be supplied as a supplement.

R= We verified assumptions for all models using hypothesis testing over model residuals in addition to visual inspection. We have included those graphs in Supplementary Material.

- Some of the predictors are known to act non-linearly such as age. As penalized regression is used, the authors are advised to include non-linear terms, such as quadratic terms.

R= We evaluated non-linear associations using restricted cubic splines and compared the performance of the model using continuous models with splines compared to categorized variables. Since we wanted the score to become widely applicable within our institution, we opted to use categorized variables; however, we have included the restricted cubic spline modeling and non-linear associations in Supplementary Materials.

- The authors should stress that the validation is internal and not external. The authors should also use test-data instead of validation data to avoid confusion when describing cross-validation (the cross-validation uses the validation data, evaluation is performed in the test data).

R= We agree with your observation, we have made that precision within the revised version of our manuscript.

- The model should be evaluated in a stratified way using the known risk groups, as this is highly relevant for practical application.

R= We performed relevant sensitivity analyses stratified by age, gender and number of comorbidities to evaluate the impact of each on the model; we have specified all conducted sensitivity analyses in Table 5 of the revised version of our manuscript.

- The authors need to provide a URL for the data set.

R= This data belongs to an internal registry for the hospital and cannot be uploaded or shared publicly. Data can be provided upon reasonable request to the corresponding author; we have uploaded R code for all performed analyses into a GitHub repository which can be verified at https://github.com/oyaxbell/nutri_cov.

---

## [Decision Letter · Decision Letter 1]

3 Dec 2020

Validation and repurposing of the MSL-COVID-19 score for prediction of severe COVID-19 using simple clinical predictors in a triage setting: The Nutri-CoV score

PONE-D-20-25115R1

Dear Dr. Aguilar-Salinas,

We’re pleased to inform you that your manuscript has been judged scientifically suitable for publication and will be formally accepted for publication once it meets all outstanding technical requirements.

Kind regards,

Itamar Ashkenazi

Academic Editor

PLOS ONE

Additional Editor Comments (optional):

Reviewers' comments:

Reviewer's Responses to Questions

**Comments to the Author**

1. If the authors have adequately addressed your comments raised in a previous round of review and you feel that this manuscript is now acceptable for publication, you may indicate that here to bypass the “Comments to the Author” section, enter your conflict of interest statement in the “Confidential to Editor” section, and submit your "Accept" recommendation.

Reviewer #1: All comments have been addressed

2. Is the manuscript technically sound, and do the data support the conclusions?

Reviewer #1: Yes

3. Has the statistical analysis been performed appropriately and rigorously? 

Reviewer #1: Yes

4. Have the authors made all data underlying the findings in their manuscript fully available?

Reviewer #1: Yes

5. Is the manuscript presented in an intelligible fashion and written in standard English?

Reviewer #1: Yes

6. Review Comments to the Author

Reviewer #1: Thank you for addressing the statistical concerns. I would just like to reiterate a concept that I including in my review of the paper. Which is that I suspect that even if this model were used at a similar institution, the efficacy observed will likely not agree with the efficacy reported in the paper.

7. PLOS authors have the option to publish the peer review history of their article (what does this mean?). If published, this will include your full peer review and any attached files.

Reviewer #1: No

---

## [Editor Report · Acceptance letter]

7 Dec 2020

PONE-D-20-25115R1 

Validation and repurposing of the MSL-COVID-19 score for prediction of severe COVID-19 using simple clinical predictors in a triage setting: The Nutri-CoV score 

Dear Dr. Aguilar-Salinas:

I'm pleased to inform you that your manuscript has been deemed suitable for publication in PLOS ONE. Congratulations! Your manuscript is now with our production department. 

Kind regards, 

on behalf of

Dr. Itamar Ashkenazi 

Academic Editor

PLOS ONE